# Wound Healing Impairment in Type 2 Diabetes Model of Leptin-Deficient Mice—A Mechanistic Systematic Review

**DOI:** 10.3390/ijms23158621

**Published:** 2022-08-03

**Authors:** Albert Stachura, Ishani Khanna, Piotr Krysiak, Wiktor Paskal, Paweł Włodarski

**Affiliations:** 1Center for Preclinical Research, Department of Methodology, Medical University of Warsaw, 02-091 Warsaw, Poland; albert.stachura@wum.edu.pl (A.S.); s079327@student.wum.edu.pl (I.K.); s082108@student.wum.edu.pl (P.K.); wiktor.paskal@wum.edu.pl (W.P.); 2Doctoral School, Medical University of Warsaw, 02-091 Warsaw, Poland

**Keywords:** diabetes, wound, leptin, wound healing, db/db, ob/ob, mice, wound healing in metabolic disease, wound healing in chronic disease, molecular mechanisms in wound healing

## Abstract

Type II diabetes mellitus (T2DM) is one of the most prevalent diseases in the world, associated with diabetic foot ulcers and impaired wound healing. There is an ongoing need for interventions effective in treating these two problems. Pre-clinical studies in this field rely on adequate animal models. However, producing such a model is near-impossible given the complex and multifactorial pathogenesis of T2DM. A leptin-deficient murine model was developed in 1959 and relies on either dysfunctional leptin (ob/ob) or a leptin receptor (db/db). Though monogenic, this model has been used in hundreds of studies, including diabetic wound healing research. In this study, we systematically summarize data from over one hundred studies, which described the mechanisms underlying wound healing impairment in this model. We briefly review the wound healing dynamics, growth factors’ dysregulation, angiogenesis, inflammation, the function of leptin and insulin, the role of advanced glycation end-products, extracellular matrix abnormalities, stem cells’ dysregulation, and the role of non-coding RNAs. Some studies investigated novel chronic diabetes wound models, based on a leptin-deficient murine model, which was also described. We also discussed the interventions studied in vivo, which passed into human clinical trials. It is our hope that this review will help plan future research.

## 1. Introduction

Within the past three decades, the number of people with diabetes mellitus has multiplied exponentially, making it the ninth major cause of death. Type II Diabetes Mellitus (T2DM) leads to impaired wound healing and hence, diabetic foot ulcers are a leading cause of non-traumatic lower-limb amputations worldwide [1,2,3]. The lifetime risk of developing a diabetic foot ulcer is between 19% and 34% [4]. A variety of therapies have been suggested for the treatment of diabetic ulcers; however, there is still an on-going search for productive treatment. This may be due to the lack of knowledge regarding the complex mechanisms involved in the development of unhealed wounds in diabetes.

Considerable resources have been devoted to developing animal models in order to research wound healing in T2DM. Among the frequently used animal models are the diabetic rodent models: db/db and ob/ob mice. Ob/ob mice lack functional leptin due to a single autosomal recessive mutation on the obese gene (chromosome 6) whereas db/db mice lack functional leptin receptors due to a single autosomal mutation on the leptin receptor gene (chromosome 4). These rodent models classify diabetes as a monogenic disease in comparison to the etiology of the human T2DM, which is polygenic as well as multifactorial in nature. Thus, even though such models are useful for research purposes, the information obtained from studies utilizing them is of limited use [5].

The animal models available for diabetes research are most often more like maturity-onset diabetes in humans. The same initial sequence of events occurs in both strains, which includes hyperphagia that is followed by a further compensatory increase in insulin secretion, and by an expansion of the Beta-cell mass. The two strains chosen for this review differ in a subsequent manifestation of the db/db mutation. The C57BL/KsJ strain presents only primary and transient hyperinsulinemia, which completely reverses and leads to insulinopenia and hyperglycemia over time. This is caused by an expansion failure of Beta cells and results in islet atrophy. C57BL/6J strain, on the other hand, manifests the mutation with islet hyperplasia that causes permanent hyperinsulinemia. These metabolic irregularities in both strains lead to obesity, specifically in the C57BL/KsJ case. Because of a raised insulin level, we can observe increased activity of gluconeogenic enzymes. Both mutants are infertile and have defects in thermoregulation. One of the main differences between C57BL/KsJ and C57BL/6J strains is the intensity of developed diabetes, a mild one in the 6J case and severe in KsJ [6,7].

In this review, we analyze the molecular mechanics of wound healing within these leptin-deficient mice models. Though the models have been in use for decades, new data accumulated gradually over time and have not been systematically described or analyzed. This article is the first to provide a comprehensive overview of wound healing biology in ob/ob and db/db mice. It also discusses different variants of wound creation (including models of chronic infected wounds) and some interventions that have been studied using this model, which passed into clinical trials. Lastly, we underline some of the key limitations and points to be considered when interpreting studies utilizing these models or planning a new experiment.

## 2. Methods

This study was conducted in accordance with the Preferred Reporting Items for Systematic Reviews and Meta-Analyses (PRISMA) guidelines [8], using a previously designed protocol (Appendix A). We searched for in vivo studies investigating leptin-deficient mice (db/db or ob/ob) with wounds or ulcers. These included studies assessing wound healing after an intervention or aimed to analyze the molecular mechanisms within the leptin-deficient mice model. All articles written in English were included, excluding reviews, letters, and editorials. Conference papers or papers without full text were excluded.

We searched using electronic databases: MEDLINE, EMBASE, Web of Science, and Scopus. To identify all relevant articles, we used prespecified search engines for each database (Appendix A). On 5 May 2021, two independent reviewers (AS, IK) conducted a systematic search of the literature.

### 2.1. Study Selection

Each relevant publication was categorized using the PICO model. Articles were included based on predefined selection criteria, or the appropriate PICO: leptin-deficient mic and db/db or ob/ob with wounds or ulcers, with any possible interventions or comparisons, assessing wound healing. Wound healing was assessed macroscopically and histologically, or molecular analyses were conducted. Exclusion criteria were human studies, review letters and editorials, inadequate PICO, and a substantial lack of methodology.

Study eligibility was assessed by screening titles and if necessary, abstracts. Later, full texts were assessed for inclusion and exclusion criteria. All disagreements were resolved by a consensus between the two reviewers.

### 2.2. Data Extraction and Analyses

Studies were divided into two categories, intervention studies and model studies, i.e., studies aiming to identify the molecular mechanisms within the leptin-deficient mice model. The following information was extracted from each intervention study by IK and PK: animal strain, wounding model (excisional, incisional, ischemic), type of intervention, and the type of wound healing outcomes (macroscopic, histological, or molecular analyses). The following information was extracted from each model study by AS: animal strain, age, and sex, as well as main outcomes. AS assessed the data extraction procedure and implemented necessary corrections.

We did not perform quantitative statistical analysis of the selected studies because of methodological heterogeneity. A systematic review of the methodology and outcomes was undertaken instead.

The primary search yielded 1433 scientific papers (Figure 1). Eventually, 105 model studies were included in this review. The most used interventions from 361 in vivo studies were summarized at the end of the literature review.

## 3. Results and Discussion

### 3.1. Literature Review

#### 3.1.1. Phenotype and Wound Healing Dynamics

Both leptin-deficient mice strains are characterized by a specific phenotype and wound healing dynamic. Animals have larger subdermal fat thicknesses and smaller dermal thicknesses. Reepithelialization plays a bigger role than contraction in wound healing [9]. Aged db/db have lower wound breaking strength and stiffness than younger db/db and healthy controls (young or aged). Hyperglycemia does not correlate with impaired wound healing [10]. Topically applied substances do not spread to adjacent wounds in db/db mice when four 6 mm wounds are created on their backs [11]. Db/db strains do not differ in their susceptibility to I/R (ischemia/reperfusion) injury from the controls [12]. Neither fasting plasma glucose nor its change during the experimental process is a significant predictor of wound closure. However, an increase in body weight significantly and independently predicts wound closure [13].

Wounds become 10% larger after excision in db/db, and 16% smaller in healthy counterparts due to contraction. One study showed that semi-occlusive dressing prolongs wound closure (27.75 days vs. 13 days in db/db). No such differences were noted for healthy counterparts. Interestingly, no statistically significant difference in wound closure rates existed between uncovered wounds in db/db and healthy counterparts in this study. There were, however, significant differences between gross epithelization and histological assessment of epithelization [11]. Contrary to the previous study, Michaels et al. showed that db/db wound healing is significantly delayed compared with the Akita strain, mice with streptozotocin (STZ)-induced diabetes, and healthy controls. Mean wound areas were larger after day 8 at all time points. Db/db also exhibited less intense granulation tissue formation than other models. There was no difference in the epithelial gap or epithelialization rate between the 3 models. Less cluster of differentiation 31+ (CD31) cells were detected on day 14 compared with the Akita and STZ-induced diabetes model. Similarly, the Ki67 index was lower in db/db than in all other models [14]. Another study by Tkalcević et al. showed slower contraction, re-epithelialization rates and wound closure, as well as less intense granulation tissue formation in db/db compared with control [15]. Another study showed significantly delayed wound closure in db/db mice, caused mainly by impaired wound contraction, not by slow reepithelialization. This finding was associated with low transforming growth factor β receptor type I (TGF-β RI) expression and attenuated small mothers against decapentaplegic 3 (SMAD3) nuclear translocation in the granulation tissue leading to impaired fibroblast-to-myofibroblast differentiation [16]. A transient increase in the wound volume appears in healthy mice due to early inflammation, which resolves during the first 7 days after wounding. In contrast, there is no such a transient increase in db/db mice due to prolonged inflammation and there is a significantly larger wound volume on days 10 and 14 post-op compared with healthy counterparts. Non-diabetic adult mice demonstrate full wound closure after 14 days, whereas in db/db animals, the wound bed shows persistent hypercellularity, indicative of prolonged inflammation, as shown by ultrasound examination [17]. Trousdale et al. found that by post-operative day 21, both wild-type and db +/− mice demonstrate complete wound closure. In db/db mice, open wounds were still present at post-operative day 21 with a range of percent wound closure from 24 to 81% with a mean of 55%. As in previous studies, no significant correlation between wound closure rate and severity of diabetes existed [18].

#### 3.1.2. Growth Factors

During wound healing, a scab is formed. The scab serves as a source of bioactive mediators, such as growth factors [19]. Endogenous growth factors are essential for normal wound repair and compared to their healthy counterparts, db/db mice exhibit dysfunctional changes in growth factors’ production (reduction and downregulation of their receptors), release, and distribution during wound healing. It prevents this process from its physiological course.

The fibroblast growth factor (FGF) family consists of at least nine different mitogens. A study investigating fibroblast growth factors found that the expression of keratinocyte growth factor (KGF) is reduced and delayed in its release in db/db mice when wounded. Basic FGF (bFGF) is released earlier, but with similar intensity in db/db mice compared to their normal counterparts. Interestingly, levels of bFGF are upregulated in the non-wounded skin. The receptors also tend to be dysfunctional in wound healing. The FGF receptor (FGFR), particularly its 2nd isoform that binds KGF and acidic FGF (aFGF), is downregulated [20]. Both platelet-derived growth factor (PDGF) and its receptor are downregulated in db/db mice. PDGF receptor A (PDFGRA) is downregulated in db/db mice in both non-wounded and wounded back skin. Furthermore, expression of the PDGFRB is also reduced during the repair process [21]. Insulin-like growth factor (IGF) I affects tissue repair whereas IGF-II influences fetal development. IGF-I is downregulated, and its production is delayed upon wounding. IGF-II peaks higher in healthy animals than in db/db mice during wound healing [22]. Hypoxia-inducible factor 1 alpha (HIF-α) plays a significant role in wound healing and hyperglycemia in db/db mice and impairs the stability of HIF-1α and vascular endothelial growth factor (VEGF). The latter enhances the permeability of local blood vessels. When destabilized, VEGF, a classical target gene for HIF, is no longer modulated by hyperglycemia as the expression of Von Hippel-Lindau (VHL) protein is downregulated. Similarly, another study found that there are decreased levels of VEGF mRNA in db/db mice during the period of granulation tissue formation [23,24]. Destabilization of HIF-1α may also lead to downregulation of its target genes. When hydroxylase inhibitors such as dimethyloxalylglycine (DMOG) and the iron chelator deferoxamine (DFX) were used, they stabilized the HIF-1α pathway, prompting the production of granulation tissue, total vessels number, and increasing cytokine receptors’ expression in endothelial precursor cells: CXC receptor 4 (CXCR4), C-Kit, Tie-2 [23]. Pigment epithelium-derived factor (PEDF) levels are elevated in type II diabetic patients with diabetic foot ulcer (DFU) as well as db/db mice. It has been proposed that overexpression of PEDF suppresses the Wnt signaling pathway in the wounded skin. When PEDF was neutralized, wound healing was accelerated, angiogenesis increased, and functions and numbers of endothelial progenitor cells (EPCs) improved [25].

#### 3.1.3. Angiogenesis

A central component of normal wound healing is angiogenesis. This process is significantly impaired in db/db mice by numerous pathological states, such as dysfunctional receptors, reduction in the number of essential ligands or cells, and epithelial to mesenchymal transition (EMT) failure (reducing the angiogenic capability).

Angiopoietins (Ang 1 through 4) target the vascular endothelium via their receptors, Tie-1 and Tie-2. Levels of Ang 1 through 4 as well as their receptors Tie-1 and Tie-2 are dysfunctional during angiogenesis in db/db mice. Both normal and db/db mice tend to have a constitutive expression of Ang-1 and upon injury, levels of Ang-2 increase. After 7 days, levels of Ang-2 begin to decrease in normal mice, however, keep on increasing in db/db mice. This overexpression of Ang-2 and the decreased levels of VEGF result in fewer endothelial cells. Similarly, Tie-1 expression is induced during wound healing and is excessively prolonged in db/db mice whereas the expression of Tie-2 is completely absent [26]. Similar to blood vessels, lymphatic vessels are necessary during wound healing. The reduction of macrophages in db/db mice may result in a significant reduction in lymphatic structures in the granulation tissue. These macrophages also show impaired recruitment via lower expressions of VEGFR3 mRNA and its ligands VEGF-C and -A. Macrophages usually express markers for lymphatic endothelium, including Lymphatic Vessel Endothelial Receptor 1 (LYVE-1), podoplanin, and Prox-1. The reduction of macrophages results in reduced lymphatic markers and hence, fewer lymphatic vessels are produced. In the normal counterpart of db/db mice, lymphatic vessels are produced de novo on day 5 after injury [27]. Wound margin keratinocytes tend to have an increase in Akt1 phosphorylation; however, this phosphorylation is dysfunctional in ob/ob mice. Insulin-mediated VEGF synthesis in keratinocytes is controlled by a Phosphoinositide 3-kinases (PI3K)/Akt/mammalian target of rapamycin (mTOR)-dependent post-transcriptional regulatory mechanism. Hence, decreased Akt1 phosphorylation will lead to poor phosphorylation of the eukaryotic initiation factor 4E-binding protein 1 (4E-BP1) and reduced levels of VEGF protein in chronic wounds of diabetic ob/ob mice. Thus, the post-transcriptional control of insulin-stimulated VEGF expression via Akt1 suggests a role of insulin in the control of keratinocyte angiogenic potential in wound healing [28]. The wound margin in db/db shows a prolonged upregulation of CXC ligans (CXCL2, CXCL5) and colony stimulating factor 3 (CSF3), TGFα, and matrix metalloproteinase 9 (MMP9), leading to a decreased expression of CD31 on days 2 and 7 after wounding [29]. Peroxisome proliferator-activated receptor γ (PPARγ) agonists improve the functions of endothelial cells and have been used as insulin sensitizers in diabetic patients. Db/db mice tend to have a reduced angiogenic potential as evidenced by isolated endothelial cells (ECs) and bone marrow-derived proangiogenic cells (PACs). This effect was partially rescued by incubation of cells with rosiglitazone (PPARγ activator). However, this effect did not manifest in vivo. Hence, db/db PACs have a decreased level of PPARγ and dysfunctional expression of PPARγ regulated genes compared with their normal counterparts [30]. Langer et al. established a model which allows high resolution in vivo imaging of functional angiogenesis in diabetic wounds compared with wild-type (WT) mice or BALB/c mice. They showed that db/db mice have a lower functional capillary density (FCD) and lower angiogenesis positive area (APA) compared with the BALB/c and WT mice [31]. Another study used a microCT analysis to visualize the 3D architecture of the capillary bed in db/db mice. They have a significantly decreased vessel surface area, branch junction number, total vessel length, and total branch number. They also have increased capillary permeability and decreased pericyte coverage of capillaries. Similarly, db/db mice tend to have a dysfunctional expression of factors associated with vascular regrowth, maturity, and stability. Specifically, the expression of VEGF-A, Sprouty2, PEDF, low-density lipoprotein receptor-related protein 6 (LRP6), Thrombospondin 1, CXCL10, CXCR3, PDGFR-β, heparin-binding EGF-like growth factor (HB-EGF), EGFR, TGF-β1, Sema-phorin3a, Neuropilin 1, angiopoietin 2, neural/glial antigen 2 (NG2), and regulator of G protein signaling 5 (RGS5) are down-regulated in diabetic wounds [32]. Hypoxia (1% O2) does not increase VEGF and heme oxygenase 1 (HO1) mRNA in db/db mice, unlike in control. HIF-1α is stable in mature db/db adipocytes in vitro, but not in vivo. This suggests that abnormal adipocyte response to hypoxia may play a role in the pathogenesis of T2DM [33].

One more molecule plays a role in diabetic angiogenesis. ZEB1 (Zinc finger E-box binding homeobox 1) is responsible for epithelial to mesenchymal transition (EMT)—a key process in wound healing—and its levels in the skin of db/db mice are particularly high. Successful depletion of ZEB1 in db/db mice improves perfusion and increases vasculature density, as well as promotes EMT. Both improved angiogenesis and EMT are associated with better wound closure. In epithelial cells, ZEB1 induces EMT toward wound reepithelialization. Hyperglycemia impairs this process. Excessive ZEB1, under diabetic conditions, may also contribute to persistent inflammation [34].

#### 3.1.4. Cytokines and Immunological Apparatus

Inflammation is the key component of wound healing, and its proper resolution is necessary for regeneration. In leptin-deficient diabetes, the course of inflammation is disturbed by an increased level of many inflammatory cytokines and elongation of their function. Upon wounding, expression of interleukin 18 (IL-18) (mainly unprocessed form) is prolonged and strongly elevated. This is associated with a longer infiltration of lymphocytes Th CD4+ and macrophages [35]. Other molecules also show prolonged expression upon wounding: IL-1β, tumor necrosis factor α (TNF-α), as well as Macrophage inflammatory protein-2 (MIP-2), and Monocyte Chemoattractant Protein-1 (MCP-1). The last two are strongly associated with a prolonged infiltration of polymorphonuclear leukocytes (PMNs) gathering below the epithelium with macrophages encircling them. CXCR2 and C-C chemokine receptor type 2 (CCR2) in turn show lower expression compared with healthy counterparts [36]. PMNs-rich wounds of ob/ob mice also show strong oncostatin M mRNA and protein upregulation, suggesting the role of this molecule in cellular infiltration. Systemic leptin administration decreases oncostatin M level, as well as PMNs count [37]. Leptin-deficient mice show low expression of IL-1β, VEGF and TNF-α compared with control, when stimulated with lipopolysaccharide + interferon γ. (LPS + IFN-γ). Nitric oxide (NO^2−^) levels, on the contrary, are higher than in control. Such a stimulus does not alter the respiratory burst in macrophages; however, they appear more rounded and clustered [38].

On day 5, inflammation decreases, and tissue healing begins. On day 8, new matrix formation is evident. There are relatively small changes between db/db and control in mRNA expression of IL-6, IL-1, MIP-2, and MCP-1. On day 8, extracellular matrix (ECM) is five-fold more abundantly present in control than in db/db mice [39]. Wound margin tissue shows strong and prolonged upregulation of CXCL2, CXCL5, CSF3, TGFα, and MMP9 [29].

Db/db mice suffer from neutrophil persistence impairing healing—treatment with Cellular Communication Network Factor 1 (CCN1) accelerates neutrophil clearance [40]. Transcription factors Forkhead Box M1 (FOXM1) and STAT3, which function to activate and promote the survival of immune cells, are inhibited in DFUs. Similar inhibition of FOXM1 is seen in db/db, resulting in delayed wound healing and decreased neutrophil and macrophage recruitment in diabetic wounds in vivo [41]. γδ T cells are normally responsible for producing cytokines and growth factors in response to damage. They become dysfunctional in ob/ob mice, as hyperglycemia disturbs their proliferation due to altered STAT5 signaling, diminishing their numbers in the epidermis. Those residing in the skin are unresponsive to epithelial cell damage due to chronic inflammatory mediators, including TNFα. Neutralizing this cytokine partially restores γδ T cells’ function [42]. Taylor et al. also showed that γδ T cells are unable to properly regulate keratinocytes’ homeostasis in obesity, which also impairs cell–cell adhesion [43].

Both leptin-deficient and healthy mice exhibit high expression of cyclooxygenase-1 (COX-1) and nearly absent COX-2 in non-wounded skin. Upon wounding, COX-1 expression declines, whereas COX-2 expression increases. On day 13 post-wounding, COX-1 expression is still reduced, and COX-2 is drastically over-expressed in db/db and ob/ob, compared with control. Systemic leptin treatment in ob/ob mice normalizes glycemia, body weight, and wound healing. It also restores COX-1 expression in wound margin keratinocytes and decreases COX-2 expression in the wound bed to levels comparable to healthy counterparts. Macrophages in the wound bed express inducible nitric oxide synthase (iNOS) and COX-2 [44]. Proper keratinocyte regeneration is associated with an increased production of prostaglandins E2 (PGE2) and D2 (PGD2), mediated by normalized COX-1 expression [45].

Upon wounding, several dysregulations occur in ob/ob mice: there are high levels of lysozyme m and lipocalin mRNA, strong long-lasting macrophage signals, and high MIP-2 levels (preceding cellular infiltration)—especially on days 5 and 7. IL-1β induced upon wounding persists over 7 days. COX-2 expression, as mentioned earlier, is markedly prolonged until at least 11 days. Importantly, even though Akt does not activate properly, nuclear factor of kappa light polypeptide gene enhancer in B-cells inhibitor, alpha (IκBα) degrades, thus further activating the nuclear factor kappa-light-chain-enhancer of activated B cells) NFκB pathway even in late wounds, prolonging inflammation [46].

Chronic inflammatory signals and endoplasmic reticulum-specific stress are associated with unfolded protein response (UPR). Wounds of ob/ob mice showed sustained induction of UPR associated with an increased expression of MIP-2. Moreover, prolongation of UPR impairs the angiogenic capacity of wound macrophages [47].

#### 3.1.5. Macrophages

In non-diabetic mice on day 5, post-injury macrophages exhibit a pro-inflammatory phenotype, manifesting itself with high expression of IL-1β, MMP9, and iNOS. This phenotype normally subsides by day 10. In db/db, however, the pro-inflammatory phenotype persists through day 10 and is associated with decreased expression of IGF-1, TGF-β1, and VEGF [48]. Such macrophages (as in humans) exhibit high expression and secretion of IL-1β. Blocking the IL-1β-mediated pathway increases the level of growth factors, as well as improves healing [49]. Rodero et al. identified a noninflammatory subset of macrophages (Ly6c^lo^MHCII^hi^), which increases gradually during normal wound healing, but is missing in ob/ob mice. Moreover, they pointed to IL-17 as a key factor inhibiting normal wound closure. Blocking IL-17 markedly improved wound healing in leptin-deficient animals [50]. Ob/ob mice also show an increased and prolonged (up until 13 days) expression of protein Ym1—a marker of the IL-4-mediated alternatively activated macrophage phenotype. Interestingly, it seems that wound macrophages take up this protein from neutrophils, which carry it to the wound bed. This represents another mechanism of macrophages’ phenotype polarisation in the healing environment, possibly contributing to inhibiting regeneration [51]. Another study presented an epigenetic mechanism skewing macrophage polarization towards the M1 phenotype and is associated with fewer wound macrophages. Hematopoietic stem cells (HSCs) derived from db/db mice exhibit markedly elevated oxidative stress levels decreasing microRNA let-7d-3p, which in turn leads to hypermethylation of genes responsible for macrophages’ differentiation [52].

Early after injury, the diabetic wound exhibits a significant delay in macrophage infiltration resulting from reduced CCL2 expression. Treatment with CCL2 stimulates healing in diabetic wounds by restoring the macrophage response. The early diabetic wound exhibits a decrease in essential macrophage response, rather than hyperactive inflammation [53]. Peritoneal macrophages exhibit impaired macropinocytosis, mediated partially by a hyperglycemia-induced decrease in AMP-activated protein kinase (AMPK) activity [54]. Macrophages isolated both from diabetic humans and db/db mice show sustained high inflammasome activity due to a lack of endogenous inflammasome inhibitors, debilitating the switch from pro-inflammatory macrophages to healing-associated phenotype [55].

#### 3.1.6. Leptin

Upon wounding, a rapid spike in leptin mRNA level is observed in db/db mice. However, leptin protein is decreased after injury, but its level comes back to normal in the period between the 3rd and the 13th day. After the injury, the functional leptin receptor subtype (obRb) declines, but comes back to normal on the 13th day. Leptin acts as a mitogen for primary epidermal keratinocytes (Figure 2) [56].

Leptin, not caloric deficit, improves wound healing [19]. It activates c-fos via the obRb leptin receptor in healthy keratinocytes, further signaling via Janus kinase (JAK) family kinases and Signal Transducer and Activator of Transcription (STAT). JAK-2 is the only kinase associated with leptin signaling. Its phosphorylation, however, differs between species and remains unclear. Protein Tyrosine Phosphatase 1B (PTP1B) is a negative regulator of leptin signaling (it dephosphorylates JAK-2). Moreover, PTP1B dephosphorylates VEGFR2, which suppresses proliferation, migration, and tube formation of vascular endothelial cells. Inhibiting this phosphatase rescues wound healing in ob/ob mice [57]. As mentioned earlier, systemic leptin treatment in ob/ob mice normalizes COX-1 and -2 expression, as well as PGE2/PGD2 biosynthesis [45]. Upon wounding and systemic leptin administration, there is a higher phosphorylation of STAT-3 (Y705) in wound tissue. This mechanism is noted both in human and murine keratinocytes [58]. Expression of VEGF protein upon injury is reduced (30 to 40%) in ob/ob mice compared with wild-type C57BL/6 animals. Systemic and topical administration of leptin reconstitutes normal wound VEGF expressions but fails to reverse the strongly reduced angiogenic response in ob/ob mice (wound vessel density) [56]. Moreover, neither systemic nor topical leptin administration induces any significant changes in hemoglobin level. It suggests that leptin accelerates wound repair by a mechanism other than stimulation of angiogenesis [59]. Immunohistochemistry confirms that the epithelium and blood vessels located in the granulation tissue express the functional leptin receptor obRb isoform during skin repair [56]. Systemic treatment of diabetic ob/ob mice with leptin blunts PMNs influx, but not macrophage influx into the wound site. Closed wounds of leptin-administered mice are characterized by tremendous numbers of macrophages within the granulation tissue. Differential effects of leptin on PMN and macrophage axes of inflammation must be indirect, as topical administration of leptin onto wounds of ob/ob mice does not reduce PMN influx into the wounded areas [19]. In the human study, from the DU group (diabetic ulcer), NDU group (not diabetic ulcer), and NC group (normal control), 10 biopsies each were examined. The cuticle thickness was significantly greater, and the epidermal layer was significantly lesser in the DU and NDU groups. Leptin protein expression was significantly higher in the DU and NDU than in the NC group (*p* < 0.001), whereas OB-RL (leptin receptor-long form) mRNA and protein expressions were significantly lower in the DU group and significantly higher in the NDU group. Diabetic foot ulcer duration was negatively correlated with OB-RL protein expression [60].

Ob/ob mice exhibit a marked overexpression of arginase-1 post-wounding with subsequent inactivity of iNOS, leading to impaired NO production. Systemic leptin treatment readjusts enzymatic expression and improves healing conditions [61].

#### 3.1.7. Insulin

In leptin-deficient mice, obesity impairs wound contraction and lowers collagen accumulation in the wound bed, regardless of the insulin treatment or diet restriction [62]. There are, however, severe dyregulations in insulin physiology in these models. Upon wounding, insulin receptor (InsR) and Glucose transporter type 4 (Glut-4) mRNA levels are significantly reduced in ob/ob mice compared with control. Protein tyrosine phosphatase 1B (PTP-1B), Insulin receptor substrate 1 (IRS-1), and IRS-2 mRNA expression profiles are not changed. However, InsR and PTP-1B protein levels decrease. Similarly, IRS-1 and IRS-2 proteins are less detectable in skin and wounds of ob/ob mice. This suggests a post-transcriptional blockade in the insulin signaling. Phosphorylated Glycogen synthase kinase-3 alpha (GSK3-α) and GSK3-β are almost completely absent in ob/ob (normally gradually increasing). No phosphorylation of GS in ob/ob is noticed, as opposed to normal mice. Glut-4 is downregulated in leptin-deficient animals. InsR is normally expressed in wound margin and in granulation tissue in healthy animals on day 5 post-wounding. On the contrary, a barely detectable signal is observed in ob/ob mice. Systemic leptin treatment restores IRS-1 and IRS-2 expression, as well as GKS3-β and GS phosphorylation, and reduces TNF-α mRNA and protein levels. Topical leptin does not improve healing, unlike systemic administration. Protein expression of the InsR, IRS-1, PTP-1B, Glut-4, and phosphorylation of GSK3-α and GS increase after leptin treatment in non-wounded skin also. This study suggests that impaired insulin signaling in obese animals is linked to increased levels of TNF-α, which in turn enhances inflammation and reduces peripheral uptake of glucose. This process is similar to the one seen in humans as human adipocytes constitutively express TNF-α [63]. Importantly, InsR downregulation plays a key role in debilitating cellular response to insulin in leptin-deficient model. This impairment, however, seems to result from a systemic, rather than local mechanism, as evidenced by no improvement after topical leptin administration [64]. It has been suggested that knockout of GM3S (ganglioside GM3 synthase), which is upregulated in ob/ob mice skin, may result in improved proliferation, migration, and activation of insulin receptor and IGF-1 receptor [65].

#### 3.1.8. Advanced Glycation End-Products (AGEs)

AGEs are heterogenous moieties endogenously produced from the glycation of proteins, lipids, and nucleic acids. Db/db mice have an accumulation of AGEs within the wound area. A high AGE concentration delays wound closure, cellularity, and inflammation at the early stages of wound healing (Figure 3). On the 14th day after wound injury, there is impaired reepithelialization, granulation, angiogenesis, and cell infiltration. The inflammation is prolonged until the 21st day post-wound injury [66]. The receptors for advanced glycation end-products (RAGE) are also dysfunctional in diabetic wound specimens. F4/80-positive macrophages and Heat shock protein 47 (HSP47)—positive fibroblasts are the major cell populations expressing RAGE during diabetic wound healing. Furthermore, diabetic wounds tend to have an increased expression of RAGE in fibroblasts and macrophages during the last stage of the repair. Thus, these results may indicate that elevated RAGE levels in diabetic wounds are associated with prolonged inflammation and deficient healing [67]. One study aimed to elucidate the skin barrier impairment in patients with T2DM and found this linked to AGEs. Murine models of diabetes presented with severe hyperglycemia, impaired skin barrier hemostasis, decreased epidermal proliferation and lipid synthesis, and decreased lamellar bodies (LB) and epidermal antimicrobial peptides (AMPs). In these animals, there is an increase in receptors for advanced glycation end-product (RAGE) in the epidermis as well as an increased serum AGE levels [68]. Db/db mice tend to have an increased expression of both RAGE and AGEs. One study attempted to support this hypothesis by blockade of RAGE using soluble RAGE (sRAGE). Administration of sRAGE accelerated the development of inflammatory cell infiltration and activation in wound foci. Cytokines such as TNF-α, IL-6, and MMP-2, -3 and -9 were suppressed. Thick, well-vascularized granulation tissue was enhanced as well as increased levels of PDGF-B and VEGF. Thus, blockade of this receptor may be a possible strategy to restore effective wound repair [69].

#### 3.1.9. Fibroblasts and Extracellular Matrix (ECM)

Fibroblasts play a pivotal role in wound healing by secreting numerous growth factors, including VEGF. Fibroblasts cultured from db/db mice have impaired migration compared with wild-type mice. Hypoxia upregulates wild-type fibroblast migration—an effect not seen in the db/db mice. Db/db-derived fibroblasts have a greater activity and concentration of MMP-9, however, no differences in MMP-2. Fibroblasts cultured from db/db mice produce 13% of normally produced VEGF—the expression not increased by hypoxia. Despite these impairments, the proliferation and senescence of fibroblasts are normal in db/db mice [70]. In this strain, a higher rate of fibroblast-specific apoptosis is seen compared with wild-type mice. This is attributed to enhanced levels of activated caspase-3 and coincides with diminished expression of collagen I and III. Hence, db/db mice tend to have a lower fibroblasts’ density and less ECM compared with wild type mice [39]. When treated with a TNF-α inhibitor (etanercept), caspase-3 levels decrease, alleviating fibroblasts’ apoptosis and increasing ECM production [71]. The pro-apoptotic transcription factor FOXO1 is excessively activated in db/db mice. Blocking TNF normalizes FOXO1 action, decreasing fibroblasts’ apoptosis, and alleviating inflammation (measured with PMNs count). In vitro analyses show that TNF-α enhances the expression of genes related to apoptosis including Akt and p53, as well as those involved in inflammation, cytokines, toll-like receptors, and NF- κB pathways [72]. Db/db mice show insufficient FGF-7 induction compared with wild-type mice, leading to a reduced wound contraction rate [73]. Substance P is an amino acid peptide, which stimulates the mobilization of bone marrow-derived mesenchymal stem cells into the bloodstream. It improves wound healing in both db/db and wild-type mice. In db/db mice, it decreases fibroblasts’ apoptosis and increases their proliferation by increasing the expressions of VEGF and stromal cell-derived factor-1 (SDF-1) [74]. Db/db-derived dermal fibroblasts have lower expression levels of IGF-1, SDF-1, connective tissue growth factor, and TGF-β. Yes-associated protein (YAP) is a mediator of mechanotransduction in dermal fibroblasts and compared to wild-type fibroblasts, db/db-derived fibroblasts have a lower expression of this protein [75].

#### 3.1.10. Apoptosis and Autophagy

Db/db mice have higher levels of apoptotic fibroblasts after a bacterial-induced wound. This is primarily due to enhanced expression of pro-apoptotic factors, such as caspase 9 (CASP9), FAS, Fas Associated Via Death Domain (FADD), and TNF-α. Compared with the wild type mice, db/db mice have a two-fold increase in CASP3 activity. CASP 3, 8, and 9 activity remains high in db/db mice even eight days after wound induction. When treated with a pan caspase inhibitor, a number of fibroblasts, expression of collagen I and III, and ECM increase [76].

Another study showed that on day 4 post-wounding, TNFα mRNA and protein levels are increased in db/db wounds. Higher caspase-3/7 activity is also detected, followed by more pronounced fibroblast apoptosis. Antagonizing TNFα enhances healing in db/db from day 5 to 9, decreasing fibroblast apoptosis. TNF-α significantly increases mRNA levels of genes involved in apoptosis by increasing apoptosis, Akt, and p53 gene sets but not mitochondrial or cell-cycle gene sets [72].

Autophagy is associated with increased light chain 3 (LC3) protein levels. Enhanced autophagy (increased LC3 levels) within db/db mice impairs the cutaneous healing process. If autophagy is inhibited by 3-Methyladenine (3-MA), proper healing is restored. The major cell type undergoing autophagy in wound healing are macrophages and it is the increased autophagy in db/db mice that induces the mobilization of macrophages. Hence, due to enhanced autophagy, db/db mice have increased expression of M1 macrophages with elevated CD11c population and gene expressions of proinflammatory cytokines. Interferon regulatory factor 8 (IRF8) is a mediator of autophagy and macrophage polarization and administration of certain AGEs results in a two-fold increase of IRF8. Subsequently, autophagic activity, and M1 macrophage polarization increase. Hence, modulating IRF8 activity could be a potential intervention for db/db mice [77].

Normally, apoptosis is limited to the wound edge and follows the advancing epithelial edge towards the center of the wound as healing progresses. Db/db mice have significant delays in the appearance of apoptotic patterns. This dysfunction is reversed after the topical application of growth factors such as IGF-II and PDGF [78].

#### 3.1.11. Stem Cells (Bone Marrow)

Stem cells also play a role in the impaired wound healing that occurs in diabetic mice (Figure 4). The following section includes a discussion of mesenchymal stem cells (MSCs), bone marrow progenitor cells (BM PCs), particularly the CXCR4-CXCL12 axis and Nicotinamide phosphoribosyltransferase—Nicotinamide adenine dinucleotide (NAMPT-NAD) pool, as well as the increased myelopoiesis in db/db mice.

Diabetic animals possess fewer MSCs than their healthy counterparts. In addition to the limited number of MSCs, their proliferation and survival are impaired. Furthermore, after wounding, significantly fewer endogenous MSCs are mobilized to the injury site. Direct application of MSCs does not accelerate wound closure in diabetic and healthy animals. However, there is a higher therapeutic outcome in healthy animals compared with diabetic animals [79].

When diabetic mice are wounded, endogenous BM-PCs are naturally mobilized. Wound re-epithelialization is the main mechanism contributing to wound closure when high levels of BM-PCs occur. The CXCR4-CXCL12 axis enhances the mobilization of diabetic BM-PCs. Treating mice with a CXCR4 antagonist increases the numbers of HSCs and EPCs in the wound bed compared with non-treated mice, as well as improves vascularization. As the CXCR4 antagonist increases, the release of these BM PCs wounds obtained from the treated animals show higher levels of phosphorylated MAPK (pMAPK), i.e., a marker of effective cell migration and activation. Further solidifying the role of the CXCR4-CXCL12 axis, when mice are treated with a c-Kit antibody (ACK2), which abrogates BM PCs mobilization, the number of HSCs and EPCs in the wound bed decreases [80].

The CXCR4 ligand—CXCL12, also known as SDF-1α, plays a role in wound closure. Unexpectedly, wild-type mice have fewer progenitor cells (PCs) than diabetic mice. However, after cutaneous wounding, the wild-type bone marrow PC number increases markedly. This does not happen in diabetic mice as the SDF-1α switch is dysfunctional. In wild-type mice, bone marrow SDF-1α increases over time as opposed to diabetic mice, where it remains constant. Thus, the mobilization of vasculogenic PCs into the circulation is impaired. Upon treating diabetic mice with plerixafor, which restores the SDF-1α switch, diabetic PC mobilization and wound closure improve [81].

Seven days post treatment with an SDF-1α inhibitor, the volume of granulation tissue decreases, similar to the number of CD31+ cells and vessel density, and the number of CD45+ cells (leukocytes) increases. There is a high expression of IL-6 in the db/db treated with the SDF-1α inhibitor on day 3 but not on day 7. However, there are higher levels of MIP-2 on day 7. Inhibition of SDF-1α also leads to a dose-dependent reduced migration of splenic leukocytes [82].

NAMPT is the rate-limiting enzyme for the synthesis of NAD. NAMPT regulates neovascularization after hind-limb ischemia by controlling the function of EPCs. It also modulates the Notch 1 intracellular domain deacetylation in a sirtuin 1 (SIRT1)-dependent manner. The concentration of NAD and the expression of NAMPT are significantly decreased in EPCs in db/db mice. When the NAD pool is enhanced, the number of circulating EPCs increases, and post-ischemic wound healing and blood reperfusion improve. Normally, SDF-1α is downregulated and endothelial NOS (eNOS) is upregulated in the bone marrow of wild-type mice. In db/db mice, this pathway works conversely. Treatment with nicotinamide and enhancing the NAD pool restores the proper pathway needed for wound healing. Additionally, enhancing the NAD pool inhibits the Poly (ADP-ribose) polymerase (PARP) and caspase-3 activates in db/db mice bone marrow. Thus, the intracellular NAMPT-NAD pool is positively associated with blood EPCs [83].

Diabetic mice have enhanced myeloid cell production after skin wounding. This is due to alterations in bone marrow progenitors present in db/db mice compared with their non-diabetic counterparts. Db/db mice have greater numbers of Lin−Sca-1+cKit+ (LSK+) cells and an increased density of MyP cells in their bone marrow compared with wild-type mice. Among the MyP subsets, only the granulocyte macrophage progenitors are significantly increased in db/db mice. Hence, the increased frequency of LSK and MyP shows that db/db mice have an increased myeloid lineage commitment at the progenitor level. The levels of mRNA for critical transcription factors and growth factor receptors associated with myeloid differentiation are upregulated in the LSK cells derived from db/db compared with WT mice. Furthermore, the expression of CSF1 receptor (CSF1R) and CSF2R alpha subunit (CSF2Rα) in the DB LSK cells is increased. Db/db mice have a significantly higher presentation of both neutrophils and Ly6Chi macrophages in the circulation and spleen, however not in the bone marrow. Thus, these findings suggest that stem cells in db/db mice may be intrinsically modified to produce myeloid lineage commitment by upregulation of critical transcription factors and receptors associated with myeloid differentiation [84].

Another study supports the claim that diabetic mice have enhanced myeloid cell production particularly due to HSCs proliferation. Cxc112 is a retention factor that promotes HSC quiescence. Endothelial cells in diabetic mice express fewer Cxc112, leading to excessive HSC proliferation. Mice with a specific endothelial-specific deletion of EGFR (Cdh5Cre Egfrfl/fl) were generated. These mice with the deletion had enhanced HSC proliferation and increased myeloid cell production. Reduced EGFR signaling decreases the expression of HSC retention factor angiopoietin-1. Inflammatory myeloid cells accumulate in skin wounds of diabetic Cdh5Cre Egfrfl/fl mice, significantly delaying wound closure. Cdh5Cre Egfrfl/fl mice also have increased atherosclerotic lesions in the aorta. Mice with hypercholesterolemia tend to have increased LSK proliferation, indicating that hypercholesterolemia may be the main driver of increased hematopoiesis in db/db mice. On the other hand, hyperglycemia may be a main driver of myelopoiesis in the STZ- and high-fat diet model [85].

Methylglyoxal (MGO) is a highly reactive dicarbonyl species that is formed during the metabolism of glucose and fructose. It converts proteins, lipids, and DNA to AGEs, which leads to cell dysfunction and organ damage. Thus, diabetic mice tend to have higher levels of methylglyoxal (MGO) due to hyperglycemia. Glyoxalase 1 (GLO1) is an MGO scavenger which reverses BMPCs dysfunction. It increases the expression of inositol-requiring enzyme 1a (IRE1a), an endoplasmic reticulum stress sensor. When diabetic mice were manipulated to augment tissue GLO1 expression, angiogenesis was enhanced, resulting in improved wound closure. Hence, GLO1 rescues BMC dysfunction and reverses MGO-induced impairment of IRE1a expression and activity [86].

#### 3.1.12. Non-Coding RNAs

MicroRNAs are small, noncoding RNAs that bind to the 3′-untranslated region and interfere with mRNA translation. Several reports have identified the role of microRNAs in the angiogenic response of wounds in db/db mice.

There is a decreased expression of miR-27b in bone marrow-derived angiogenic cells (BMAC) of db/db mice, leading to dysfunctional angiogenesis. Normally, miR-27b is required to enhance BMAC function. Firstly, miR-27b suppresses antiangiogenic molecules such as Thrombospondin-1 (TSP-1) and Semaphorin 6A (Sema 6A). TSP-1 is an extracellular glycoprotein required to modulate cell-to-cell interactions. Hyperglycemia decreases the expression of miR-27b leading to increased expression of TSP-1. Secondly, miR-27b suppresses p66shc, which is a pro-oxidant protein and hence, contributes to the protection of bone marrow aspirate concentrate (BMAC) function by preventing mitochondrial oxidative stress. When miR-27b is given to db/db mice, BMAC function improves, including proliferation, adhesion, tube formation, and delayed apoptosis, and hence, partly improved wound healing [87]. Another study revealed that miR-26a is induced in db/db mice due to the increased concentration of glucose in endothelial cells. MiR-26a in the wounds of db/db mice is elevated compared with WT mice. Inhibiting miR-26a increases angiogenesis, and granulation tissue thickness and accelerates wound thickness. MiR-26a enhances bone morphogenetic protein (BMP)/SMAD1 signaling by decreasing the expression of SMAD1 and its downstream regulator inhibitor of DNA binding 1 (ID1), and further on, increases the expression of cell cycle inhibitor p27. The use of the miR-26a inhibitor reverses this effect [88]. Db/db mice also have reduced levels of miR-132, which plays an important role in inhibiting inflammation and promoting the growth of epidermal keratinocytes. TGF-β plays a major role in wound healing and regulates the expression of miR-132; when dysfunctional, it reduces miR-132 levels. miR-132 also suppresses the NF-κB signaling pathway and affects other inflammation-related signaling pathways, including the NOD-like receptor, Toll-like receptor, and TNF signaling pathways [89]. Db/db mice also have an increased expression of miR-615-5p within their wounds. It inhibits the VEGF-Akt/eNOS signaling pathway in endothelial cells. Specifically, miR-615-5p targets IGF2 and the Ras-association domain family 2 (RASSF2) which decrease the VEGF-AKT/eNOS signaling, impairing angiogenesis. Furthermore, miR-615–5p inhibits the release of nitric oxide, which drives endothelial cell angiogenic responses. Endothelial cell proliferation and migration are reduced in db/db mice. Neutralization of the microRNA by local delivery of its inhibitors increases angiogenesis and granulation tissue thickness enhancing wound closure [90].

#### 3.1.13. Bacterial Burden and Biofilm

On average, 40 times more bacteria colonize db/db mice skin than healthy counterparts, which is also associated with a greater variety of species. The genera Staphylococcus and Aerococcus are present more abundantly, whereas Streptococcus, Lachnospiraceae, Incertae, and Sedis are scarce in db/db skin as compared with db/+ skin. Upon wounding, the bacterial load increases, and the species’ colonizing skin changes. Gene transcripts were compared between db/db and control with a punch biopsy at baseline and subsequent days, showing little differences at day 0. Later, however, the gene expression seems to be delayed in db/db mice compared with the control group. This finding is correlated with the selective Staphylococcal shift [91]. Biofilm within the wound promotes chronicity as bacteria express genes associated with cellular growth, thus exhausting oxygen resources in the wound bed. Additionally, they promote immune cells’ migration to the wound site. These in turn use oxygen for defense-related processes (such as oxidative burst performed by PMNs). Overall, these effects lead to diminishing oxygen tensions and a prolonged healing duration [92]. Importantly, a state of lasting hypoxia is present in db/db mice even without any bacterial burden. Though it is difficult to assess it in an excisional wound (as measurements are highly influenced by atmospheric oxygen), a more accurate measurement is possible in pedicled flaps [93].

Db/db mice have an impaired ability to clear the wound of bacteria at days 5 and 10. Additionally, 12 h after inoculation, the db/db inflammatory response is greater than in healthy animals, and they also exhibit a higher myeloperoxidase (MPO) activity, which normalizes by 24 h. The early inflammatory response in db/db is non-effective as the bactericidal activity of neutrophils is reduced, an effect not mediated directly by hyperglycemia. PMNs present reduced respiratory bursts. Interestingly, bone-marrow derived neutrophils showed no difference in bactericidal activity between the groups [94].

Some changes in the dermal connective tissue are noted between db/db and control mice upon bacterial infiltration. On day 1, inflammation develops in both db/db and control with comparable TNF-α levels between the groups. On day 3, inflammation is prolonged in db/db with more PMNs, whereas it resolves in control. At the same time, the expression of TNF-α, MIP-1, and MCP-1 is prolonged in the former group [95].

#### 3.1.14. Models of Chronic Diabetic Wounds

Some authors attempted to modify the original leptin-deficient murine model to reflect more accurately clinical situations such as infected diabetic wounds (Table 1). Zhao et al. challenged the wounds with P. aeruginosa biofilm two days post-wounding. This resulted in prolonged epithelialization, more extensive inflammation, tissue necrosis, and epidermal hyperplasia adjacent to wounds. The model proved reproducible and had low mortality rates [96,97]. Their follow-up study showed deficient vascularization in the same model, with HIF expression markedly elevated. Such wounds heal in approximately 6 weeks [98]. Later, Chen et al. showed that inoculating DB/db wounds with P. aeruginosa increased the ratio of pro-inflammatory cytokines to anti-inflammatory cytokines, and increased numbers of both M1 and M2 macrophages with greater wound persistence in the former group [99].

Another group challenged db/db mice wounds with excess oxidative stress by blocking catalase and glutathione peroxidase. After such an intervention, the biofilm developed within the wound spontaneously, rendering the wound chronic. Treatment with antioxidants α-tocopherol and N-acetylcysteine reduced oxidative stress, and the biofilms had increased sensitivity to antibiotics; furthermore, granulation tissue formation and morphology were restored [100]. Another group used the same method to create a protocol for chronic diabetic wounds. They showed that such an approach could prolong healing to as long as 60 days. Here, however, they used db/db mice on BKS background and the animals were 5–6 months old [101]. Panayi et al. argued that blocking antioxidant enzymes results in high mortality, especially in older animals. They decided to use modified doses of catalase and glutathione peroxidase inhibitors in 47 11-week-old db/db mice and noted no deaths after this procedure. Wounds challenged in such a way showed an arrested inflammatory state during healing [102].

As contraction plays an important role in wound regeneration in rodents, Park et al. investigated the effect of splinting on the healing dynamics. They found that silicone splint application reduced wound contraction in heterozygous and db/db mice, making the process more similar to that in humans, in which wounds heal primarily by granulation [103].

#### 3.1.15. Other Studies

The outcomes of several studies do not fit into any of the previous chapters, thus we summarized them here.

NEP (neutral endopeptidase) degrades substance P—a neuropeptide stimulating proliferation and migration of keratinocytes. The activity of NEP is usually increased in patients with chronic wounds, as it is elevated is in unwounded skin of db/db mice [104]. HMG-CoA reductase (HMGR) is responsible for keratinocyte angiogenic and proliferative responses, found mainly at the wound margin. Its mRNA expression peaks between day 3 and 5, as well as on day 13. This biphasic expression is not present in ob/ob mice, possibly contributing to wound healing impairment [105]. P38 is phosphorylated (peaks) on days 1 through 7 after wounding in db/db mice, possibly accelerating inflammatory processes. Blocking this protein results in accelerated wound healing (reduced wound width, accelerated re-epithelialization, increased granulation, and reduced inflammatory cell infiltration into the wound) [106].

One group investigated wound healing in db/db mice exposed to healthy peripheral circulation. They performed parabiosis at week 4 and wounding at week 8. Wound healing enhanced by parabiosis with WT animals showed an increased granulation tissue amount and more collagen deposition. Platelet endothelial cell adhesion molecule-1 (PECAM-1) and Ki-67 signals increased in chimeras, compared with db/db control. The blood exchange rate in anastomoses also improved. Cell proliferation increased, though not reaching the level of WT controls. Interestingly, WT fibrocytes may be recruited into the wound from WT circulation and facilitate ECM deposition. Vessel formation, however, relies mainly on db/db cells. Approximately 20% of total cells in the wound area were derived from WT animals in chimeras. Parabiosis did not influence glycemic control. Improvement in wound healing was also observable when animals were separated 24 h after wound formation [107]. Skin flap tissue re-integration is also impaired under diabetic conditions in ob/ob mice. Flap tissue loses early VEGF expression, as well as wound margin keratinocytes, associated with reduced blood vessel formation. Importantly, HIF-1α is completely absent in diabetic flap tissue. These findings suggest difficult conditions that may appear when performing reconstructive surgeries under diabetic conditions [108]. Surprisingly, when creating a myocutaneous flap in ob/ob or db/db mice, increased perfusion and revascularization rates are observed, coupled with reduced flap necrosis on day 10, compared with healthy littermates [109].

One group showed that topical application of 11β-hydroxysteroid dehydrogenase (11β-HSD1) enhances wound healing in ob/ob mice, also proving that 11β-HSD1 negatively regulates the proliferation of keratinocytes and fibroblasts [110]. A conflicting study showed that 11β-HSD1 levels in intact db/db murine skin are in fact elevated and argued that it may contribute to impaired wound healing in this model [111]. The issue of corticosteroids’ synthesis in the skin under diabetic conditions seems to be unresolved. Another study showed that systemic and topical administration of adiponectin improves wound healing in db/db mice, augmenting proliferation and migration of keratinocytes via AdipoR1/AdipoR2 and the Extracellular signal-regulated kinase (ERK) signaling pathways [112].

Sood et al. aimed to assess the metabolic profile of intact and wounded skin of db/db mice 7 days after injury. They found 62 metabolites, which showed an altered response to injury compared with healthy counterparts. It was the first study pointing to glycine, kynurenate, and OH-phenylpyruvate being dysregulated in the wound healing process [113]. Zhao et al. investigated the role of Epoxyeicosatrienoic acids (EETs)—molecules, whose production is catalyzed by cytochrome P450. Ob/ob mice showed lower mRNA and protein expression of cytochrome P450 2C65 and 2J6 (CYP2C65 and CYP2J6), suggesting low EETs levels. Furthermore, 11,12-EET partially rescued wound healing by modulating inflammation and angiogenesis, posing a potential therapeutic agents in leptin-deficient model [114].

#### 3.1.16. Interventions

Innumerable publications have attempted to find an intervention that would be viable in wound healing in the leptin-deficient mice model. However, only a handful of these has passed the preclinical phase and entered human clinical trials (Table 2).

PDGF had shown success in pre-clinical studies and its recombinant form, Becaplermin (REGRANEX^®^ Gel) [115], was approved for the topical treatment for lower extremity diabetic neuropathic ulcers that extend beyond the subcutaneous tissue. Unfortunately, in 2008, a black box warning was added to the gel as there had been an increased rate of deaths from systemic malignancies seen in patients who received three or more tubes [116]. Beyond 2008, post-marketing studies have demonstrated no increased safety risk with the use of the gel [117]. Other interventions, which successfully proceeded to clinical trials are displayed in Table 2.

**Table 2 ijms-23-08621-t002:** Interventions studied in leptin-deficient murine models, which progressed to clinical trials.

Intervention	In Vivo Studies	Most Advanced Clinical Trial	Reference
rPDGF-BB (Becaplermin)	[118,119,120,121,122,123,124]	Phase 3	[125]
Collagen application (scaffold)	[126,127]	Phase 4	[128]
Chitosan Gel Application	[129,130,131]	Phase 2	[132]
bFGF	[119,133,134]	Phase 3	[135]
VEGF, HIF1-α	[136,137,138,139]	Phase 2	[140]
Negative Pressure Wound Therapy	[141]	Phase 2	[142]
Cold Plasma Therapy	[143,144]	Phase 2	[145]
Topical Insulin	[111,146]	Phase 2	[147]
Low Magnitude High-Frequency Vibration Platform	[148]	Phase 2	[149]
Nitric Oxide Releasing Patch	[150,151,152]	Phase 3	[153]

rPDGF-BB—recombinant human BB homodimeric platelet-derived growth factor; bFGF—basic fibroblast growth factor, VEGF—vascular endothelial growth factor, HIF1-α—hypoxia-inducible factor 1 alph.

### 3.2. Discussion

The study intended to comprehensively summarize both the mechanistic and functional aspects of using T2DM murine models of leptin deficiency in wound healing research. The wound healing process relies on numerous molecular and physiological pathways. Thus, the coexistence of DM modulates the process. A vast majority of analyzed papers focused on deepening the knowledge of shifts in molecular pathways as either a systemic or local response to injury. The most prevalently described pathologies in wound healing concerned: extension in complete wound healing time in both excisional and incisional settings, impaired growth factor expression especially via distorted HIF1α pathway, severely impaired angiogenesis at various stages, distorted macrophage role, prolonged proinflammatory cytokine excretion, and impaired apoptosis. Along with local discrepancies, also systemic alterations are present, such as distorted stem cells function.

Conclusions drawn from the summary are limited by a few factors. The study aimed to address only the leptin-deficient murine model of T2DM. Thus, it lacks findings concerning diabetic wound healing on other rodent models such as Zucker diabetic Sprague Dawley rats, Zuker fatty rat, SHR/N-cp Rat, or JCR/LA-cp Rat [5,154]. In addition, the review did not cover dietary or pharmacologically induced T2DM models—such C57BL/6J mice fed with a high-fat diet [155] or low-dose streptazocin-treated animals [156]. However, small size, cost efficiency, obesity and hyperglycemia, easy availability of those mice, widespread use in wound healing research, and delayed healing are still advantages of the leptin-deficient murine model. The above-mentioned are counterbalanced with unfavourable traits such as loose and hairy skin, wound healing with contraction (overcome by wound splinting), difficulty in inducing partial thickness wounds, limited translational efficiency, and an immune response distinct from that of humans [154]. Additionally, the background strain of leptin-deficient mice influences the phenotype. C57BLKS background leads to an uncontrolled rise in blood sugar, severe depletion of the insulin-producing beta-cells of the pancreatic islets, and death by 10 months of age [157]. Mice with the C57BL/6 background have compensatory hyperplasia of the islet B cells, and continued hyperinsulinemia throughout an 18- to 20-month life span. Eventually, wound healing is delayed and metabolic efficiency is increased [7]. Thus, researchers should be aware of those details when designing a study or analyzing the literature.

Some important distinctions must be underlined between the human and murine skin anatomy, physiology, and molecular signaling. Despite the same layers of cells in the dermis and epidermis, human skin is relatively thick (over 100 μm) and adhered to the underlying tissues, whereas murine skin is thinner than 25 μm and loose. Human epidermis is composed of 5 to 10 cell layers, whereas murine skin contains only 2 or 3. Murine dermis is thicker and 40% firmer in males compared to females, whereas the epidermis and subcutaneous tissue are thicker in the latter. Further, panniculus carnosus found in murine subcutaneous tissue is absent in humans, and thus influences skin biomechanics during wound healing—up to 90% of excisional wounds in mice close by contraction. Histologically, epidermal ridges are present in human skin and absent in mice. However, collagen types I and III, respectively, are similarly distributed in mice and human skin. There are also several differences in skin immunology. The phenotypic markers of macrophages differ between species. F4/80 adhesive glycoprotein, for instance, identifies both murine macrophages and human eosinophils. Conversely, the macrophage mannose receptor (CD206) found in murine M2 macrophages is expressed in human dermal fibroblasts and keratinocytes. Langerhans and CD8-positive T cells populate the human epidermis. Murine epidermis, in addition to these cell types, contains a specific population of γd dendritic epidermal T cells (DETCs). They secrete FGF-9 in injured skin and promote the WNT pathway, upregulating FGF-9 in dermal fibroblasts, finally leading to hair follicle regeneration. Such a mechanism could explain hair follicles formation in mice scars but not in humans. They play an exceptional role in murine wound healing, unlike in humans. Further, IL-8, CXCL-7, CXCL-11, and monocyte chemoattractant were identified in humans but not in mice wound healing. Human skin also specifically expresses CCL-13, CCL-14, CCL-15, and CCL-18, whereas CCL-6, CCL-9, CXCL-15, CXCL-14, and CCL-12 are only expressed in mice [158].

Noteworthy, the monogenic origin of db/db and ob/ob mice do not correspond with the polygenic nature of T2DM in humans. Leptin deficiency is not the main causative mechanism of T2DM. Thus, considering polygenic models such as NONcNZO10 seems worthwhile. Compared with db/db and STZ-induced DM models, only NONcNZO10 preserved impaired wound healing phenotypes in different wound models compared with the healthy origin strain. Db/db were inefficient in ischemia-reperfusion lesions [12].

The multifactorial nature of the wound healing process becomes even more complex with coexisting diabetes. The aforementioned studies, despite their limitations, serve as an important preclinical basis for translational applications. In total, 27 studies served as in vivo preclinical studies, prior to further clinical trials (*n* = 10). The most promising results concerned rPDGF-BB (Becaplermin), Collagen application (scaffold), Trafermin (bFGF spray), and Nitric Oxide Releasing Patch which achieved phase III clinical trial.

The discrepancy between the number of in vivo intervention studies (*n* = 361) and clinical trials confirms the inevitable need for use of the animal model in wound healing studies—both basic and applicatory ones.

## 4. Conclusions

An impressive number of studies relies on db/db or ob/ob mice as T2DM models for wound healing in both basic and applicatory research. Despite their limitations hindering the transnationality of results into the human setting, they remain reliable and widely studied animal models. Their strength is based on a diligently studied mechanisms of wound healing on both local and systemic levels along with satisfactory reflection on pathologies present in human diabetic wounds. Mechanistic studies revealed some degree of resemblance and discrepancy between the animal models and human physiology. With some precaution, especially concerning the monogenic nature of the model, they may be further used in preclinical studies of wound healing for incisional and excisional lesions with comorbid diabetes mellitus.

## Figures and Tables

**Figure 1 ijms-23-08621-f001:**
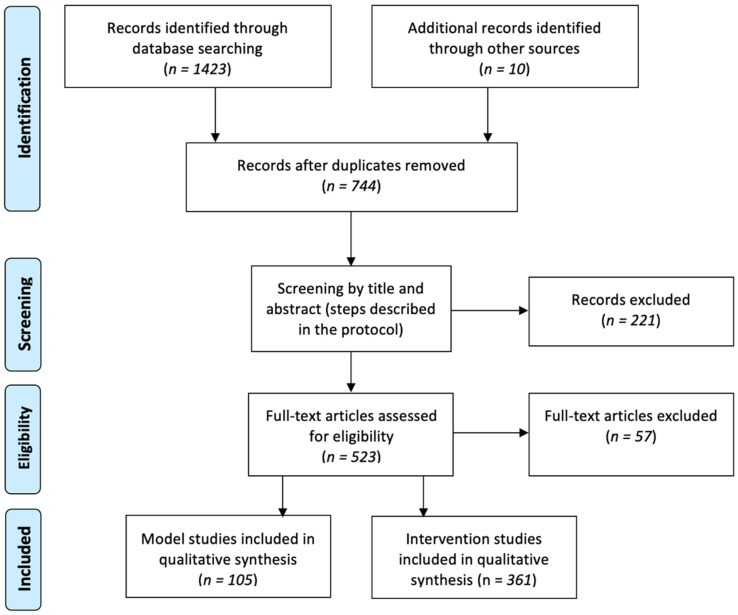
PRISMA flowchart of article selection.

**Figure 2 ijms-23-08621-f002:**
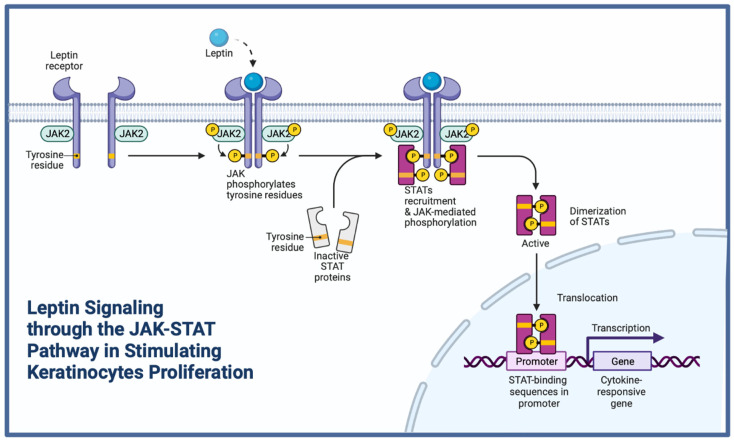
Leptin acting as a mitogen in wound healing. JAK-STAT—Janus kinase-signal transduction and transcription activation.

**Figure 3 ijms-23-08621-f003:**
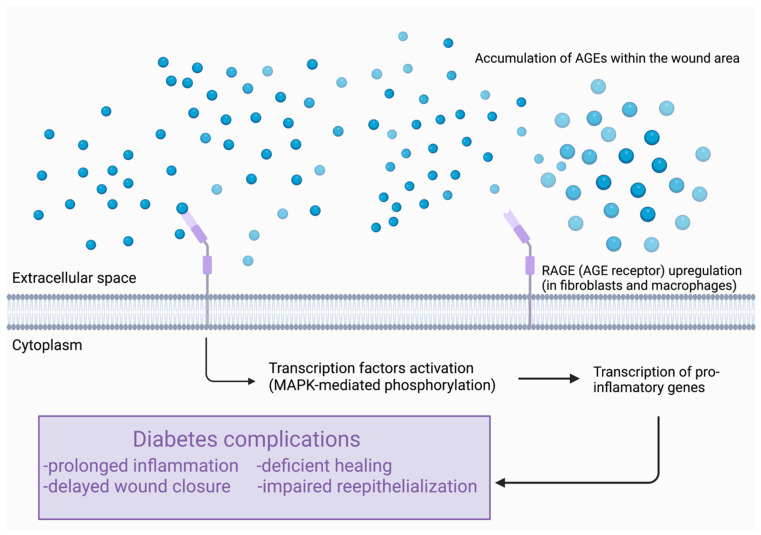
Effect of AGEs (advanced glycation end-products) on wound healing process. MAPK—mitogen-activated protein kinase.

**Figure 4 ijms-23-08621-f004:**
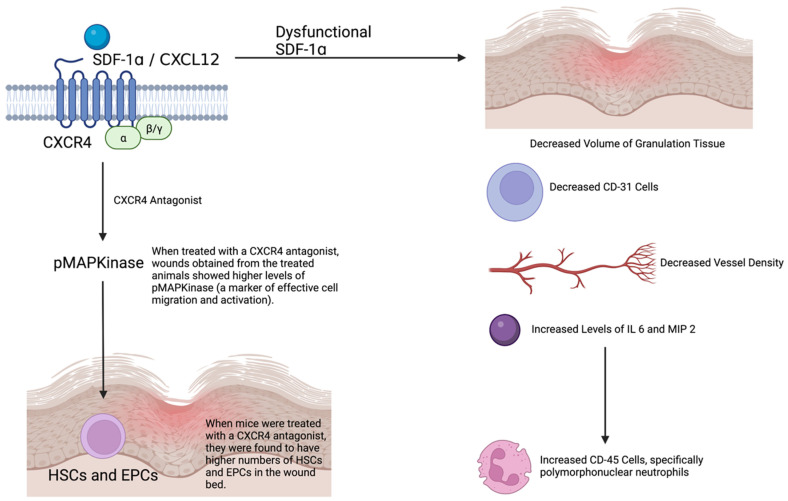
CXCR4/CXCL12-dependent stem cell function in wound healing. HSCs—hemapoietic stem cells, EPCs—endothelial progenitor cells, pMAPK—phosphorylated mitogen-activated protein kinase, CXCR—CXC chemokine receptor, SDF—stromal cell-derived factor, CXCL—CXC chemokine ligand, CD—cluster of differentiation, IL-6—interleukin 6, MIP 2—macrophage inflammatory protein 2.

**Table 1 ijms-23-08621-t001:** Modifications of leptin-deficient mice models.

Animal Model	Modification	Ref.
db/db (BKS) 11 weeks old, females	Pseudomonas aeruginosa biofilm challenge	[97,98]
db/db (C57BL/6)/db/db (BKS) 5–6 months old/db/db 11 weeks old, males	Increasing oxidative stress via blocking catalase and glutathione peroxidase	[100,101,102]
db/db (BKS) 12 weeks old, males	Wound splinting to prevent contraction	[103]

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
