# Peer review of "Wound Healing Impairment in Type 2 Diabetes Model of Leptin-Deficient Mice—A Mechanistic Systematic Review"

_ijms, 2022, doi:10.3390/ijms23158621_

Round 1

Reviewer 1 Report

The present manuscript presents a comprehensive summary of wound healing in the type 2 diabetes model of leptin-deficient mice. The advantage is an in-depth summary of current results detected in the pathophysiology of wound healing solely in these mice. Although information may serve as a good basis for understanding the model peculiarities, several issues deserve further addressing.

Comments

1. The manuscript is presented as pseudo-original research with methods, results, and discussion. This is misleading because the authors just explain the selection of studies for review without any statistics or additional analysis. Further text is a standard review. Reordering of the manuscript is therefore recommended. 

2. Manuscript is focused only on mouse leptin models without specification of their limits in comparison with human situations. It is known that wound healing in mice follows different patterns than in humans such as contraction etc. These differences should be outlined with clear specifications of advantages and disadvantages. 

3. The rationale of these models is to study the pathology of wound healing with limited value due to interspecies differences, but especially to test topical or systemic therapeutic agents. However, the responsiveness of the model to different pharmacotherapy is not properly described together with corresponding human data which confirms the validity of these models for testing the agents. Just a few agents were mentioned in chapter 3.17. but again without a comparison of results.

4. Corresponding rat models exist such as ZDF with a mutation in leptin receptor, which is also frequently used for wound studies. Are there any differences between these rat and mice models? 

5. Organization of chapters such as 3.3. 3.4. and many others is confusing. Starting with a list of numerous details such as factors, and cytokines, without explaining their functions and significance for wound healing. A crucial point is frequently hidden in the text and the reader is confused. E.g. chapter 3.4. the crucial information about the reduced angiogenic potential in these mice is in the middle of the chapter. The reorganization of chapters outlining the crucial messages will be beneficial - like e.g. chapter 3.9. Figures may help in this context.

6. English should be checked - especially if the discussion contains several typing errors - eg. line 726 mechanistc, 729 coexisteence, 730 deepeening.

Reviewer 2 Report

The authors declare a systematic Review regarding Wound healing impairment in diabetes type 2 model of leptin- deficient mice – a mechanistic systematic review. My major suggestions related to this manuscript are bellow, as follows:

Please check and apply the Instructions for authors regarding: the way for inserting references in the text (brackets not superscript).

Please avoid the personal manner of addressing “we” and our”, replacing it with the impersonal one. The text will sound more professional. Please revise the entire manuscript in this regard (i.e. L60, L65)

L32. Why naming USA? Many recent studies, published in MDPI as well, are reported regarding diabetic foo syndrome all over the world (I suggest checking and referring to more recent papers than 2005 https://doi.org/10.3390/medicina56080380)

Aim of the study L60-67. The topic is not a new one, of course. Please provide what novel/special aspects your study provides, as to underline better the necessity of it.

As being a Review, you can name the sections as you best consider for their content. For the 2nd section, I suggest replacing the actual title with Methodology of literature selection or something similar. The content of the section is good.

3rd section is wrongly named. As I mentioned above, the titles of the sections/subsections must reflect the content. You have NO real results. All is other’s results. And this is a Review, NOT an original article, to have these sections , named as you have done.

3.1. belongs to the 2nd section but I suggest removing it as it duplicates the information already provided in Figure 1.

Extend the figures on the entire width of the page. They will be readable!!!! For the same reason, in the figures, use letters with similar size of the main text. Moreover, under each figure, after the title of the figure, please explain all abbreviations used on that figure (checking Instructions for authors).

Also, ALL abbreviations used in the text MUST be explained at their first insertion. According to the same Instructions for authors “Acronyms/Abbreviations/Initialisms should be defined the first time they appear in each of three sections: the abstract; the main text; under the first figure or tablewhen defined for the first time, the acronym/abbreviation/initialism should be added in parentheses after the written-out form.” Revise the entire manuscript in this regard.

3.8. subsection is too poor developed. Please proceed accordingly.

Table 2, Reference column must contain the reference(s), not the link.

I suggest detailing in a separate chapter the potential interaction between leptin and endothelial dysfunction mediated by insulin-resistance and the consequences of endothelial dysfunction on the capacity of wound healing (https://doi.org/10.3892/etm.2020.9327 ). Also describe potential ways of treatment of PAD peripheral artery disease in the context of leptin mediated atherosclerosis, such as cilostazol, or other anti-aggregates. You can make a figure at the end of the article where to summarize the effect of leptin deficiency on all the components/factors of wound healing.

Round 2

Reviewer 2 Report

I do not see the relevance of db and ob as keywords.

Figure 2. The mouse and the triangle are huge vs the text on the figure. Why that triangle occupy 1/5 of the figure, with no significance, an the text is so small?

References must be completed by providing all the information requested in the Instructions for authors. If you EndNote them or using Mendeley Reference manager, is much easier to chose the MDPI style for writing the refs.
